# How to Create a Regional Diatom-Based Index: Demonstration from the Yuqiao Reservoir Watershed, China

**Yufei Liu** [1,2], **Jing Fang** [2,*], **Pengyu Mei** [3], **Shuo Yang** [2], **Bo Zhang** [4] **and Xueqiang Lu** [1,*]

1   Tianjin Key Laboratory of Environmental Technology for Complex Trans-Media Pollution, Tianjin International Joint Research Center for Environmental Biogeochemical Technology, College of Environmental Science and Engineering, Nankai University, Tianjin 300350, China
2   School of Geography and Environmental Science, Tianjin Normal University, Tianjin 300387, China
3   Tianjin Eco-Environmental Monitoring Center, Tianjin 300191, China
4   R&D Department, FS Limited, Katikati 3129, New Zealand
*   Correspondence: jq19066818jingxi@163.com (J.F.); luxq@nankai.edu.cn (X.L.)

**Abstract:** Diatom-based indices derived from the percentage of diatom taxa groups can be used to assess water quality. As some diatoms are location-dependent, such diatom indices are correspondingly location-dependent and the regional classification of taxa group is thereby needed. This study aims to demonstrate how to create a regional diatom assemblage index (YRDAI) based on a case study from the Yuqiao Reservoir watershed, China. Herein, we proposed a simple taxa classification approach based on the correlation between pollutant concentration and diatom abundance, and compared it with the traditional approach based on the coexistence index. Using the two approaches, a total of 34 diatom taxa groups were reclassified for localization of the well-known DAIpo index. The YRDAI was then derived from the reclassified diatom taxa groups. The results in the Yuqiao Reservoir watershed showed that the correlation-based YRDAI$_{cor}$ scores could better reflect the pollution levels of COD and TP than the coexistence-based YRDAI$_{co}$ scores and the original DAIpo scores with the unreclassified taxa groups. It can be expected that the precision of YRDAI can be improved with the accumulation of the diatom data, and the above approaches can be applied to other watersheds for making their own regional indices.

**Keywords:** regional diatom assemblage index; YRDAI; DAIpo; water quality

## 1. Introduction

With the rapid growth of the population and intensification of industrial and agricultural activities, the freshwater ecosystem has been damaged continuously [1]. To understand the health status of water bodies, water quality can be evaluated by scoring the physical, chemical and biological indices of water bodies based on expert knowledge and government standards [2,3]. Another way to evaluate the water health status is to assess the community characteristics of aquatic organisms through biological monitoring. For example, since the EU Water Framework Directive (WFD) was initiated, all EU states were required to use biological indicators to monitor and evaluate water quality [4]. Because some aquatic organisms are location-dependent, the location-specific knowledge on the relationships between aquatic organisms and water indices are crucial for application of the method. However, such work in many developing countries such as China started later than the developed countries, although some scholars have studied this topic [5–8]. There is still a gap of knowledge on the relationship between biological indicators and environmental variables.

A diatom-based index, as one of the biological indicators for assessing water quality, has become increasingly popular, because a diatom-based index has been proven to be effective even in the streams where the concentration of a pollutant in water can change significantly within a few hours [9,10]. Meanwhile, the diatom-based index is convenient,

as only a small amount of sediment samples is required to quantitatively describe benthic diatom communities [11,12]. In most cases, diatoms respond to changes in water chemistry with a delay, therefore, they reflect relatively long-term status in water chemistry [13–15]. For aquatic ecosystems characterized by high stress, diatom-based indices performed better than other biological indices, in terms of reflecting pollution gradients and impacts of specific pollution sources [16].

At present, there are various diatom-based indices, which can reflect the status of organic pollution, eutrophication or specific pollutants of water bodies. For example, the diatom assemblage index (DAIpo) developed by Watanabe et al. was used to reflect the organic pollution status of rivers in Japan [17], and this index has also been applied in East and Southeast Asia [18–21]. For the DAIpo index, species were divided into three groups, saprophilous, saproxenous and indifferent taxa groups, according to the sensitivity of different species to biological oxygen demand and the symbiosis between species. The DAIpo score was derived from the percentages of three taxa groups [17,22]. The trophic diatom index (TDI) developed by Kelly for assessing river eutrophication is widely used around the world, especially in Europe [23–26]. For the TDI index, species were allocated to one of five groups based on their sensitivity to phosphorus. Following the similar approach of the TDI, Oeding and Taffs developed a regional diatom index, the Richmond river diatom index (RRDI), for evaluating the eutrophication status of sub-tropical Australian rivers [12]. Other diatom indices for evaluating specific pollutants, e.g., herbicides and pesticides, were also developed based on the difference of diatom sensitivity to pollutants [27].

Diatom-based indices were regarded as having significant applicability across geographic regions due to the cosmopolitan nature of species [28,29]. Diatom-based indices were mainly developed in Japan, Europe or North America [30–34]. Despite a lack of information on the ecological preference and tolerance of diatoms in some countries, diatom-based indices were often directly used [21,35]. However, the environmental selection process was the major driver for diatom biogeography [36,37], and because of the regional differences in associated taxonomic variability within the indices [29,38,39], diatom-based indices responded inconsistently to measured environmental variables in different regions. The development of a new diatom index requires a long-term series of data. Instead, some researchers attempted to create the "borrowed" diatom index through localization of an existing diatom-based index, (e.g., Oeding and Taffs, 2017). Nevertheless, localization of the diatom-based index is necessary to achieve better evaluation results than the direct use of the initial index [40,41], based on the hypothesis that diatoms are location-dependent.

In this study, a correlation-based approach was proposed to replace the coexistence-based approach for simplifying the procedure of diatom reclassification, which is crucial in creating a regional diatom assemblage index. The Yuqiao Reservoir watershed was chosen as an example to validate the approach and demonstrate the localization of DAIpo, a diatom-based index that originated from Japan.

## 2. Methods

### 2.1. Establishment of the Yuqiao Reservoir Watershed Diatom Assemblage Index (YRDAI)

2.1.1. General Routine for YRDAI Establishment

For establishment of a regional diatom index, it is crucial to reclassify the diatom taxa groups based on the relationship between the diatom community and environmental pollutants. The previous approach, the coexistence index of diatom, can be used to reclassify the taxa groups through updating the regional data. Meanwhile, we proposed a new approach based on the correlation between diatom taxa and pollutants. Figure 1 shows the procedure to establish the regional diatom assemblage index for the Yuqiao Reservoir watershed (YRDAI) from the "borrowed" diatom assemblage index (DAIpo).

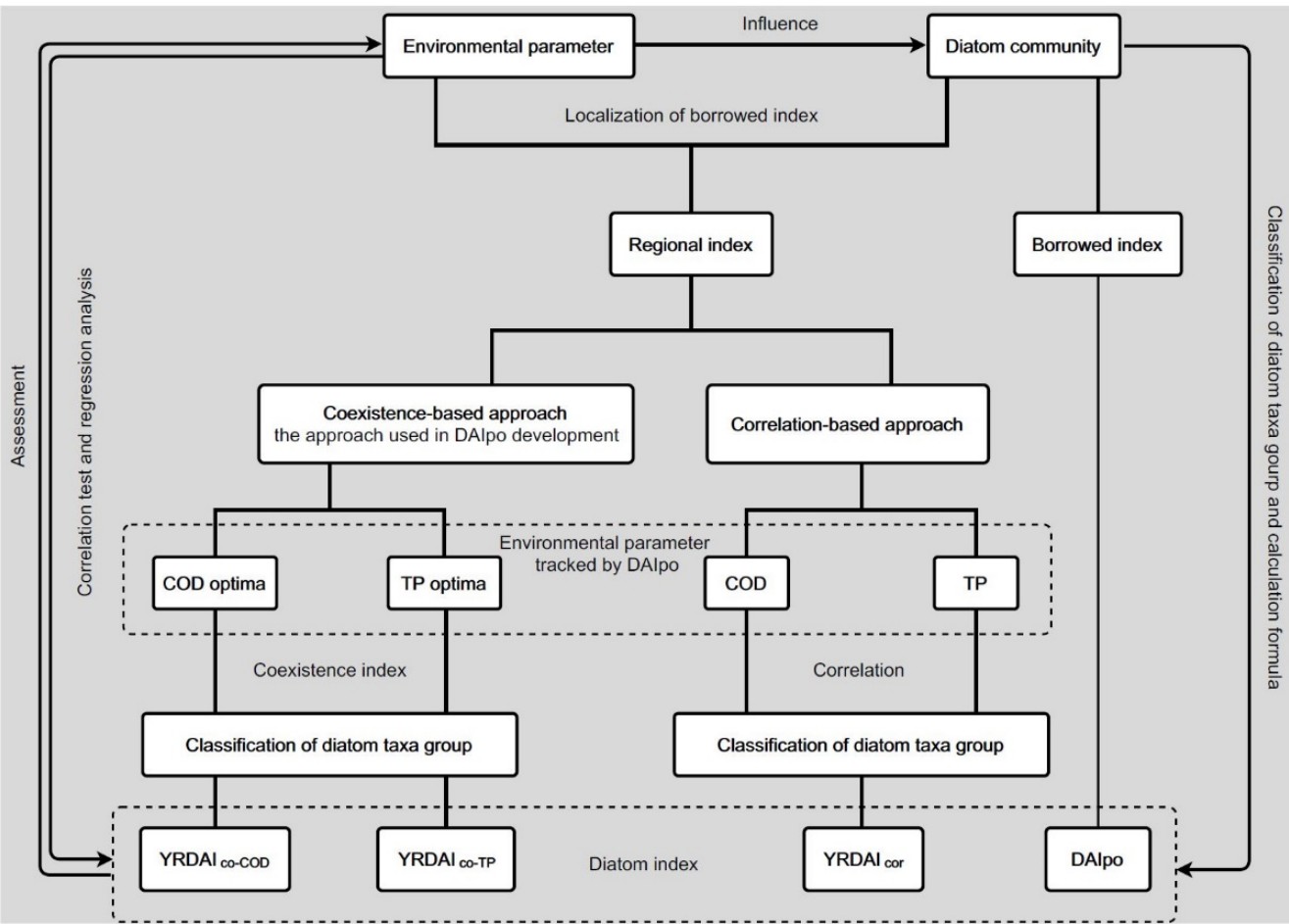

**Figure 1.** Flow chart for creating a regional diatom index (YRDAI).

Compared with other environmental factors, such as pH and electrical conductivity, organic pollution and eutrophication have a greater impact on diatom growth [42]. Organic pollution can be characterized by BOD or COD, and eutrophication is related to nitrogen and phosphorus in water. DAIpo was initially constructed based on the relationship between organic pollution and the diatom community, which also had a correlation with total nitrogen (TN) and total phosphorus (TP). According to the regional water quality standard (GB3838-2002) in the study area, three commonly used parameters (COD, TN and TP) were selected to establish the regional diatom index. However, the diatoms seemed not to be TN-sensitive in this study. Therefore, COD and TP were used to reclassify the diatom taxa groups when developing YRDAI.

### 2.1.2. Coexistence-Based Approach for Diatom Reclassification

The coexistence index approach followed Watanabe (2005) with a few modifications [43]. The pollutants were selected from pollution characteristics and the relationship with diatom taxa. Pollutant concentration thresholds were then determined. In this study, the thresholds for COD and TP were determined as {<detection limit, $\geq$40 mg/L} and {$\leq$0.02 mg/L, $\geq$0.3 mg/L}, respectively, based on the Chinese National Surface Water Quality Standard (GB3838-3002). The pollutant optima value of diatoms was derived from the tolerance index, and the diatom species was classified into three ecological groups (saprophilous species, indifferent species and saproxenous species) by the coexistence index.

The tolerance index (TI) for diatom species that occurred in the study area was computed using diatom proportion as its weight, namely:

$$TI_j = \sum_{i=1}^{n} Pollutant_i \times P_{ij} \tag{1}$$

where $TI_j$ is the tolerance index value of species $j$; $Pollutant_i$ is the pollutant concentration at sampling site $i$; $Pij$ is the proportion of species $j$ at sampling site $i$; $n$ is the total number of sampling sites. Here, $TI$ is also the optima value for the diatom species.

The coexistence index (CI) between species $j$ and species $k$ ($CI_{jk}$) was calculated using the following equation developed by Pianka (1973) [44]:

$$CI_{jk} = \frac{\sum_{i=1}^{n} P_{ij} \times P_{ik}}{\left( \sum_{i=1}^{n} P_{ij}^2 \times \sum_{i=1}^{n} P_{ik}^2 \right)^{\frac{1}{2}}} \tag{2}$$

A total of 25 benthic diatom taxa were reclassified using this approach, and each species was classified into saprophilous, saproxenous or indifferent taxa groups.

### 2.1.3. Correlation-Based Approach for Diatom Reclassification

The proportion of diatoms with high pollutant sensitivity in a diatom community gradually decreased with the increase in pollutant concentration [43,45]. Therefore, we proposed a simple method to classify diatom groups through the correlation between the proportion of each diatom species and the pollutant concentration at each sampling site. The positive, negative and noncorrelations indicate the diatom belongs to the saproxenous, saprophilous and indifferent taxa group, respectively.

### 2.1.4. Calculation of Diatom Index Score

The calculation of YRDAI score followed the DAIpo score [43], and the index score ranges from 0 (very serious pollution) to 100 (very slight pollution). According to the two classification approaches above, four diatom indices, i.e., YRDAI$_{co-COD}$, YRDAI$_{co-TP}$, YRDAI$_{cor-COD}$ and YRDAI$_{cor-TP}$ were calculated. YRDAI$_{co-COD}$ and YRDAI$_{co-TP}$ were the coexistence-based indices obtained from the COD and TP tolerance sequences for reclassified diatom taxa groups, respectively. YRDAI$_{cor-COD}$ and YRDAI$_{cor-TP}$ were the correlation-based indices derived from the reclassified diatom taxa groups through the correlation analysis between diatom species and the two pollutants (COD, TP), respectively. Because the reclassification results from two pollutants were the same, YRDAI$_{cor-COD}$ and YRDAI$_{cor-TP}$ were also the same. Therefore, YRDAI$_{cor}$ was used to represent YRDAI$_{cor-COD}$ and YRDAI$_{cor-TP}$ for convenience.

## 2.2. Case Study Area and Sample Analysis

### 2.2.1. Yuqiao Reservoir

The watershed of Yuqiao Reservoir with a total area of 2060 km$^2$ is located in North China (39°56′~40°23′ N, 117°26′~118°12′ E), where the climate is temperate continental monsoon climate. The three inflow rivers are the Li River (st.1–st.6), the Sha River (st.7–st.26) and the Lin River (st.27–st.33), and average annual runoff is 506 million m$^3$. The areas of wood-grass land and farmland in the watershed of Yuqiao Reservoir account for 54.22% and 31.81%, respectively. The three rivers receive all nonpoint source pollution from farmland and orchards, and point source pollution [46].

### 2.2.2. Sample Collection and Analysis

Thirty-two sampling sites were set up in the watershed of Yuqiao Reservoir. Sampling was conducted during the normal water season in mid-May 2019 (Figure 2). Water samples (500 mL) were taken from each site using plastic bottles, then stored in an incubator at 4 °C.

Surface sediment (50 g for each) was collected by a grab and brought back to the laboratory in a sealed plastic bag.

**Figure 2.** Study area and sampling locations.

Preparation of sediment samples: A total of 2 cm$^3$ of each sample was put into a conical flask, and 15% hydrogen peroxide was added to remove the organic matter. After the reaction was sufficient, clean water was added. The upper suspension was poured out after standing for 24 h, and then clean water was filled again. The upper suspension was poured out again after standing for 4 h. This operation was repeated for 4 times. Preparation of diatom slides: The heating plate was heated to 180 °C, then slide and cover slips were placed on a heating plate and heated to keep them dry. After the sample was shaken for 10 s, a small amount of sample was dropped on the cover slip using a dropper. Then, heated neutral balsam was dropped on the slide, and dried cover slips were covered on neutral balsam. The slide was then moved from the heating plate to operating table and put into a sealed box after cooling. More than 400 diatoms were identified at each sampling site, except the sites (st.11, 12, 24 and 26) with a small number of diatoms.

For the pollutant analysis, COD was measured using dichromate method, TP was measured using ammonium molybdate spectrophotometric method and TN was measured using alkaline potassium persulfate digestion by ultraviolet spectrophotometry.

*2.3. Data Analysis*

SPSS Statistics 19 (IBM, Armonk, NY, USA) and Origin 2018 (OriginLab, Northampton, MA, USA) were used for statistical analysis to explore the relationships between the environmental variables and the diatom data. All water quality variable measurements below detection limits were expressed as half the detection limit for statistical purposes [12]. The regression and correlations were performed using Pearson's product-moment correlation coefficient.

**3. Results**

*3.1. Pollutants*

Water quality analysis showed an increasing pollution gradient from the Lin River basin with weak human activities to the upper reaches of Li River and Sha River with a higher degree of urbanization (Table 1). The area with relatively high pollution was concentrated in the vicinity of Zunhua city. In areas with a high degree of urbanization (st.1, 2, 3, 4, 10, 11, 12 and 15), the average value of COD was 27.25 ± 14.28 mg/L, the average value of TP was 0.22 ± 0.18 mg/L, and the average value of TN was 7.96 ± 4.33 mg/L. In the Lin River basin (st.27, 28, 29, 30, 31 and 32) where human activities were relatively

weak, the average value of COD was 2.67 $\pm$ 1.49 mg/L, the average value of TP was 0.03 $\pm$ 0.03 mg/L and the average value of TN was 6.73 $\pm$ 3.96 mg/L. It is worthy to note that COD value at st.5 (84 mg/L) was abnormally high. We checked the historical water quality for the nearest routine monitoring station. The routine monitoring results showed that the maximum value of COD concentration was 30 mg/L during 2015–2019. The water sample of st.5 was obviously affected by accidental or occasional factors, and the data for st.5 were therefore excluded from further analysis to explore the relationship between water parameters and diatom communities.

**Table 1.** Pollutants and diatom index scores in the Yuqiao Reservoir watershed, China.

| Site | COD (mg/L) | TN (mg/L) | TP (mg/L) | DAIpo | YRDAI$_{co-COD}$ | YRDAI$_{co-TP}$ | YRDAI$_{cor}$ |
|---|---|---|---|---|---|---|---|
| 1 | 41 | 5.01 | 0.09 | 39.98 | 12.42 | 9.91 | 52.83 |
| 2 | 15 | 5.69 | 0.3 | 49.32 | 46.02 | 49.32 | 49.32 |
| 3 | 4 | 12.5 | 0.04 | 48.74 | 30.96 | 30.96 | 30.96 |
| 4 | 34 | 6.56 | 0.14 | 41.89 | 33.99 | 40.57 | 46.38 |
| 5 | 84 | 4.48 | 1.13 | 77.56 | 71.71 | 73.66 | 75.61 |
| 6 | 8 | 5.74 | 0.04 | 82.73 | 42.58 | 49.76 | 58.39 |
| 7 | 6 | 2.9 | 0.03 | 55.88 | 39.21 | 50.00 | 58.75 |
| 8 | 40 | 3.36 | 0.09 | 60.66 | 29.27 | 41.33 | 43.09 |
| 9 | 6 | 3.94 | 0.03 | 62.32 | 44.95 | 38.50 | 59.04 |
| 10 | 14 | 7.22 | 0.06 | 52.64 | 40.77 | 52.16 | 52.16 |
| 11 | 37 | 17.3 | 0.61 | 11.96 | 5.26 | 11.96 | 11.96 |
| 12 | 48 | 3.22 | 0.21 | 52.48 | 16.58 | 46.29 | 46.29 |
| 13 | 28 | 8.57 | 0.17 | 66.54 | 50.12 | 50.25 | 64.44 |
| 14 | 29 | 5.54 | 0.48 | 57.84 | 47.35 | 36.53 | 54.64 |
| 15 | 25 | 6.16 | 0.34 | 61.41 | 61.41 | 28.16 | 59.80 |
| 16 | 2 | 10.5 | 0.01 | 79.20 | 64.11 | 70.56 | 73.72 |
| 17 | 7 | 4.28 | 0.05 | 66.11 | 62.32 | 66.11 | 66.11 |
| 18 | 2 | 4.71 | 0.01 | 80.16 | 50.46 | 52.75 | 60.78 |
| 19 | 2 | 2.16 | 0.01 | 59.84 | 48.76 | 49.77 | 73.19 |
| 20 | 2 | 3.79 | 0.005 | 52.23 | 49.63 | 64.48 | 70.05 |
| 21 | 2 | 3.27 | 0.01 | 70.03 | 39.80 | 50.63 | 55.79 |
| 22 | 4 | 5.98 | 0.04 | 52.02 | 49.76 | 54.29 | 85.36 |
| 23 | 2 | 4.5 | 0.01 | 51.29 | 49.53 | 49.65 | 94.85 |
| 24 | 10 | 9.22 | 0.01 | 75.81 | 44.01 | 50.23 | 64.52 |
| 25 | 13 | 6.03 | 0.36 | 52.71 | 24.17 | 16.75 | 53.07 |
| 26 | 16 | 8.67 | 0.43 | 65.97 | 46.99 | 34.03 | 57.18 |
| 27 | 2 | 3.77 | 0.09 | 87.00 | 68.56 | 86.64 | 97.52 |
| 28 | 2 | 4.55 | 0.01 | 53.07 | 50.00 | 53.07 | 92.38 |
| 29 | 2 | 4.85 | 0.01 | 67.44 | 49.27 | 49.27 | 79.88 |
| 30 | 2 | 4.57 | 0.02 | 66.18 | 50.42 | 50.42 | 75.63 |
| 31 | 2 | 15.2 | 0.02 | 69.06 | 47.40 | 52.72 | 81.68 |
| 32 | 6 | 7.43 | 0.02 | 56.36 | 46.09 | 70.78 | 54.28 |

*3.2. Diatoms*

The diatom species identified from the three main rivers in the Yuqiao Reservoir watershed included 136 species from 35 genera (Figure 3). In the 136 species recorded, 25 and 21 species were reclassified by coexistence- and correlation-based approaches, respectively (Table 2).

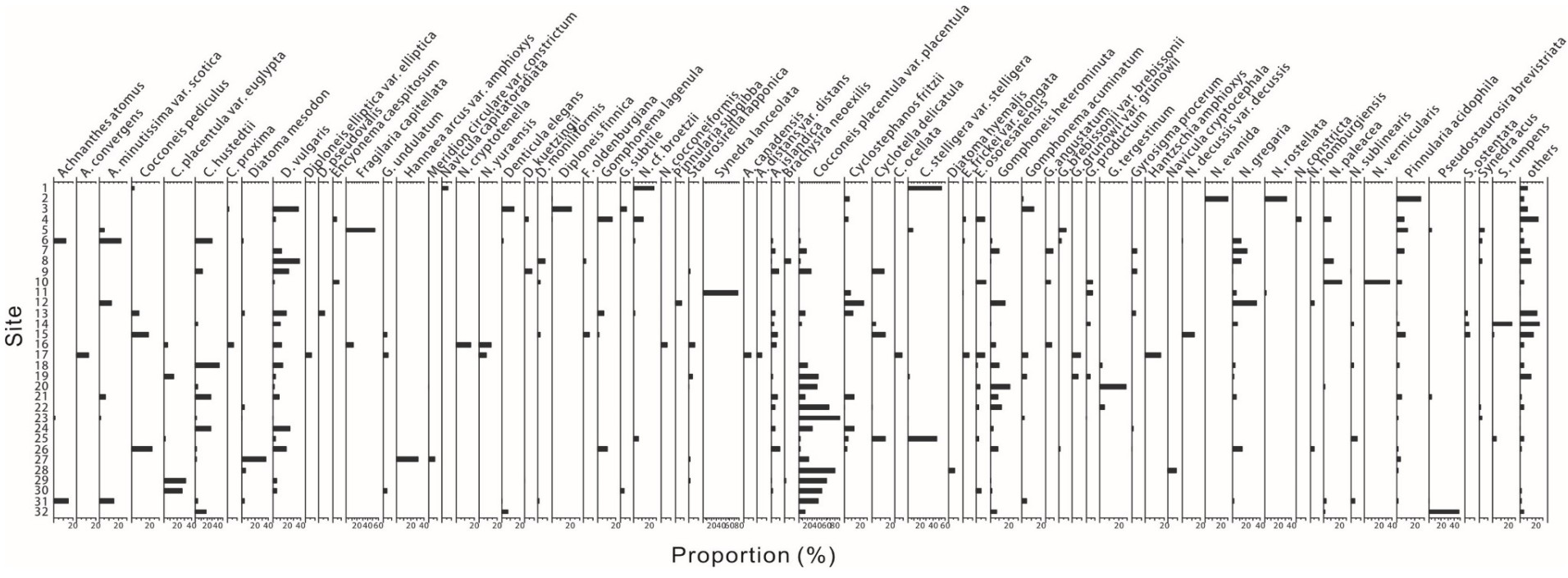

**Figure 3.** Diatom distribution for each sampling site. All the diatom species with a relatively low percentage are included in the category entitled as Others.

**Table 2.** Diatom TI value and taxa groups in the Yuqiao Reservoir watershed.

| Taxon | Abbreviation | Origin Group [43] | TP | | | COD | | | Correlation | |
|---|---|---|---|---|---|---|---|---|---|---|
| | | | Order | Optima (mg/L) | Group | Order | Optima (mg/L) | Group | | Group |
| *Achnanthes atomus* | AATO | saproxenous | 7 | 0.03 | indifferent | 9 | 4.73 | indifferent | | |
| *A. minutissima* var. *scotica* | ASCO | saproxenous | 10 | 0.05 | indifferent | 12 | 10.27 | indifferent | non | indifferent |
| *Aulacoseira islandica* | AISL | indifferent | | | | | | | non | indifferent |
| *Cocconeis pediculus* | CPED | saproxenous | 18 | 0.31 | saprophilous | | | | | |
| *C. placentula* var. *euglypta* | CEUG | saproxenous | 6 | 0.03 | indifferent | 5 | 2.37 | indifferent | non | indifferent |
| *C. placentula* var. *placentula* | CPLA | indifferent | 8 | 0.03 | indifferent | 8 | 3.95 | indifferent | negative | saproxenous |
| *Ctenophora pulchella* | CPUL | saproxenous | | | | | | | non | indifferent |
| *Cyclostephanos fritzii* | CFRI | indifferent | 14 | 0.19 | indifferent | 15 | 23.02 | saprophilous | non | indifferent |
| *Cyclotella delicatula* | CDEL | indifferent | 17 | 0.27 | saprophilous | | | | non | indifferent |
| *C. stelligera* var. *stelligera* | CSTE | indifferent | 15 | 0.20 | saprophilous | 17 | 28.14 | saprophilous | | |
| *Cymbella hustedtii* | CHUS | saproxenous | 9 | 0.03 | indifferent | 10 | 5.32 | indifferent | non | indifferent |
| *Diatoma mesodon* | DMES | saproxenous | | | | 7 | 3.94 | indifferent | non | indifferent |
| *D. vulgaris* | DVUL | saproxenous | 12 | 0.10 | indifferent | 13 | 14.99 | indifferent | non | indifferent |
| *Eunotia osoresanensis* | EOSO | indifferent | | | | | | | non | indifferent |
| *Fragilaria capitellata* | FCAP | saproxenous | 1 | 0.01 | saproxenous | 1 | 2.00 | saproxenous | | |
| *Gomphoneis heterominuta* | GHET | indifferent | 11 | 0.05 | indifferent | 11 | 8.02 | indifferent | non | indifferent |
| *Gomphonema acuminatum* | GACU | indifferent | | | | | | | non | indifferent |
| *G. lagenula* | GLAG | saprophilous | 16 | 0.20 | saprophilous | | | | | |
| *G. productum* | GPRO | indifferent | | | | | | | non | indifferent |
| *G. tergestinum* | GTER | indifferent | 3 | 0.01 | saproxenous | 4 | 2.28 | indifferent | | |
| *Gyrosigma procerum* | GPRO | indifferent | | | | | | | non | indifferent |
| *Hannaea arcus* var. *amphioxys* | HAMP | saproxenous | | | | 2 | 2.00 | saproxenous | | |
| *Navicula* cf. *broetzii* | NBRO | saprophilous | | | | 18 | 34.84 | saprophilous | non | indifferent |
| *N. cryptotenella* | NCRY | saproxenous | 2 | 0.01 | saproxenous | 3 | 2.00 | saproxenous | | |
| *N. decussis* var. *decussis* | NDEC | indifferent | 19 | 0.33 | saprophilous | | | | | |
| *N. gregaria* | NGRE | indifferent | 13 | 0.15 | indifferent | 14 | 21.37 | saprophilous | non | indifferent |
| *N. sublinearis* | NSUB | indifferent | | | | | | | non | indifferent |
| *N. yuraensis* | NYUR | saproxenous | 5 | 0.03 | saproxenous | 6 | 3.88 | indifferent | | |
| *Nitzschia paleacea* | NPAL | indifferent | | | | 16 | 23.35 | saprophilous | non | indifferent |
| *Pinnularia acidophila* | PACI | indifferent | | | | | | | non | indifferent |
| *Pseudostaurosira brevistriata* | PBRE | indifferent | 4 | 0.02 | saproxenous | | | | | |
| *Staurosirella lapponica* | SLAP | saprophilous | | | | | | | non | indifferent |
| *Synedra lanceolata* | SLAN | saprophilous | 21 | 0.61 | saprophilous | | | | | |
| *S. rumpens* | SRUM | indifferent | 20 | 0.46 | saprophilous | | | | | |

For the coexistence-based approach, the pollutant optima values for the 25 diatoms were calculated, and the diatoms were ranked with their TI values. Two boundaries based on the COD concentrations were recognized at the order between *Navicula cryptotenella* and *Cocconeis placentula var. euglypta* and between *Diatoma vulgaris* and *Navicula gregaria*, and the other two boundaries based on the TP concentrations were recognized at the order between *Navicula yuraensis* and *Cocconeis placentula var. euglypta* and between *Cyclostephanos fritzii* and *Cyclotella stelligera var. stelligera* (Figure 4). We classified these diatoms as saproxenous or saprophilous taxa groups, and the diatoms in the middle of the order sequence were classified as indifferent taxa group (Table 2).

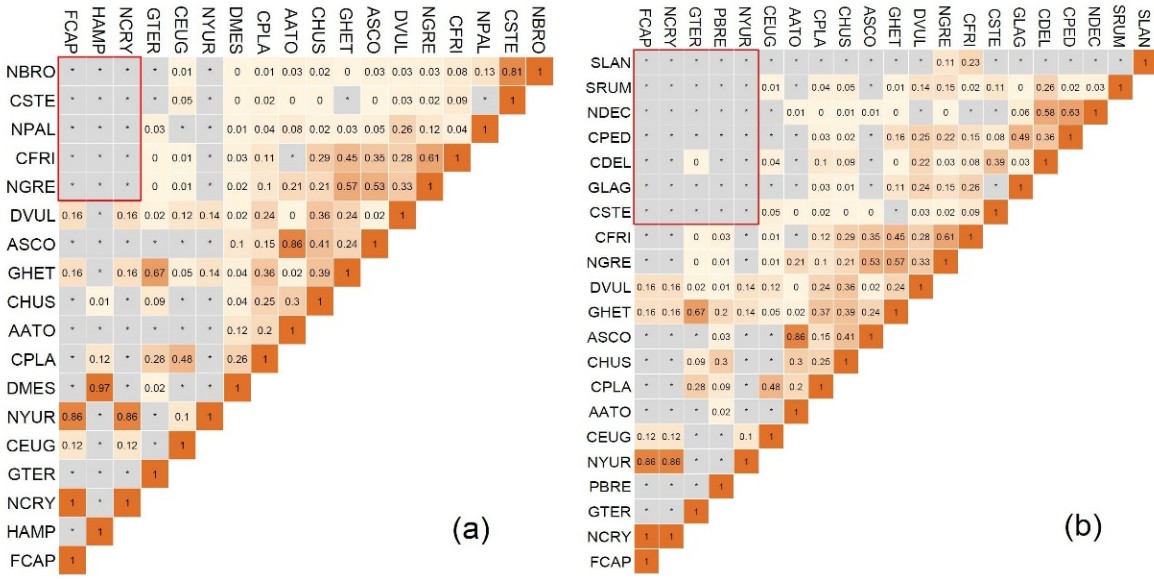

**Figure 4.** Matrix of coexistence index of each pair of diatom species which were selected by the coexistence-based approach, and the diatoms were ordered according to TI values based on (**a**) COD and (**b**) TP. Grey color means that two diatoms were not found at the same sampling location. The abbreviations of species are listed in Table 2.

The correlation results showed that only the proportion of *Cocconeis placentula var. placentula* had a significant negative correlation with the COD (r = −0.56, *p* < 0.01) and TP (r = −0.50, *p* < 0.05) concentrations. Therefore, we classified it into the saproxenous taxa group, and the remaining 20 diatom species were classified into the indifferent taxa group (Table 2).

*3.3. YRDAI Performance*

The YRDAI$_{co-COD}$, YRDAI$_{co-TP}$ and YRDAI$_{cor}$ scores ranged (5.26, 68.56), (9.91, 86.64) and (11.96, 97.52), respectively. DAIpo scores using the unreclassified diatom groups, ranged from 11.96 to 87. The lowest scores of DAIpo, YRDAI$_{co-COD}$ and YRDAI$_{cor}$ appeared near Zunhua City, while the lowest score of YRDAI$_{co-TP}$ appeared in st.1 in the upper reaches of Li River, where residential areas and factories were gathered.

The regression analysis between diatom indices and pollutant concentrations can indicate the performance effect of the indices (Table 3, Figure 5). Both the YRDAI$_{co}$ (for COD and TP) and YRDAI$_{cor}$ scores had a higher correlation with the pollutants (COD and TP) than the DAIpo scores, indicating that the regional diatom assemblage index (YRDAI) showed the improved performance compared the original diatom assemblage index (DAIpo). For the YRDAI, the YRDAI$_{co-COD}$ and YRDAI$_{cor}$ had the best performance for COD and TP, respectively.

**Table 3.** Results of correlation analysis quantifying the significance of the relationship between diatom indices and total phosphorus (TP), total nitrogen (TN) and chemical oxygen demand (COD). The symbols * and ** indicate significant correlation at the level of 0.05 and 0.01, respectively.

| | COD | TN | TP | DAIpo | YRDAI$_{cor}$ | YRDAI$_{co\text{-}COD}$ | YRDAI$_{co\text{-}TP}$ |
|---|---|---|---|---|---|---|---|
| **COD** | 1 | | | | | | |
| **TN** | 0.09 | 1 | | | | | |
| **TP** | 0.58 ** | 0.35 | 1 | | | | |
| **DAIpo** | −0.49 ** | −0.28 | −0.47 ** | 1 | | | |
| **YRDAI$_{cor}$** | −0.61 ** | −0.36 * | −0.54 ** | 0.55 ** | 1 | | |
| **YRDAI$_{co\text{-}COD}$** | −0.65 ** | −0.24 | −0.37 * | 0.70 ** | 0.72 ** | 1 | |
| **YRDAI$_{co\text{-}TP}$** | −0.57 ** | −0.27 | −0.58 ** | 0.61 ** | 0.62 ** | 0.73 ** | 1 |

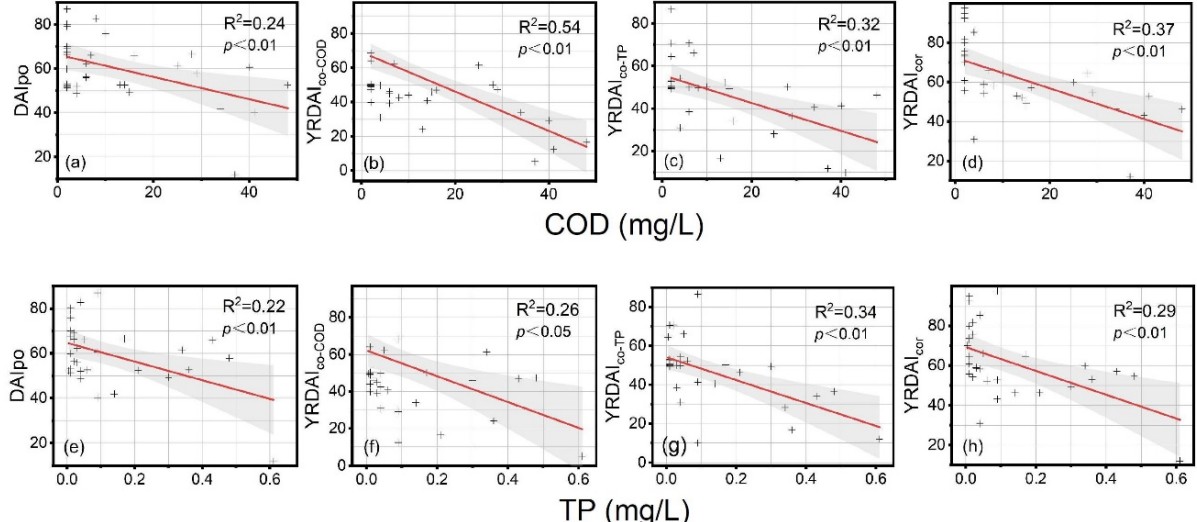

**Figure 5.** Relationship between COD (**a**–**d**) and TP (**e**–**h**) concentrations and the diatom indices with 95% confidence interval.

As shown in Figure 6, there were obvious differences in the scores from the two approaches. Generally, the scores from the correlation-based approach were greater than the scores from the coexistence-based approach. For the coexistence-based approach, the scores of YRDAI$_{co\text{-}TP}$ were slightly greater than those of YRDAI$_{co\text{-}COD}$.

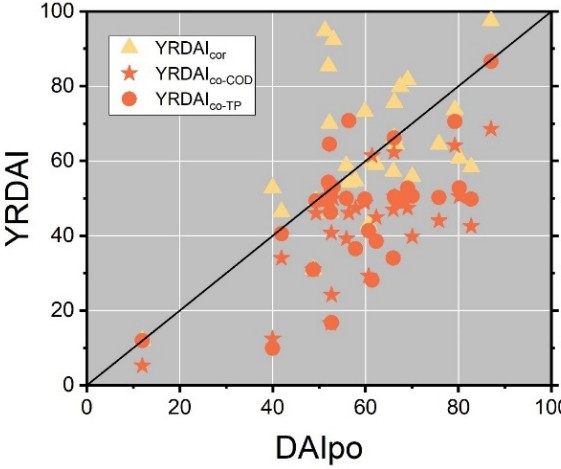

**Figure 6.** Comparison of DAIpo, YRDAIco and YRDAIcor scores.

## 4. Discussion

The diatom-based index may not reflect all of the pollutants. In other words, the pollutants should be verified and selected when constructing a regional diatom index. In this study, TN was not used to reclassify the diatom species, because the large variance of TN led to chaotic order of TI values and a weak correlation with the proportion of diatom species. The high variance of TN may be attributed to the complex forms (nitrate, nitrite, ammonia and organic nitrogen) and environmental behaviors (e.g., nitrification, denitrification and anammox processes), and possible various sources. In addition, DAIpo is originally a diatom index related to organic pollution ((BOD) biological oxygen demand), although it can also reflect other pollutants in many cases [21,47,48].

The key to the establishment of a diatom index is to classify the ecological preferences of dominant species, and the development of a regional diatom index based on the existing diatom index often followed the classification method of the original index [12,27]. However, lack of knowledge of the ecology of diatoms in the studied watershed may lead to the wrong classification of diatoms and increase the uncertainty of the developed regional diatom index [12,20,27]. When diatoms with small proportions are used to calculate the index score, the calculated results will be severely affected, resulting in an inaccurate assessment of water quality [49]. Therefore, we selected the diatoms with a proportion of >10% for at least one sampling site as representative diatoms for classification, in order to avoid the bias caused by the species with low abundance. Although the sparse diatoms should be excluded, an increase in diatom species number may improve the confidence in the diatom index score for water quality assessment.

In total, 34 diatom species were reclassified using the two approaches (Table 2). The reclassification of diatom species for YRDAI$_{co-COD}$ and YRDAI$_{co-TP}$ increased the saprophilous taxa, which led to the lower scores of YRDAIco-COD and YRDAIco-TP than that of DAIpo. In the case of YRDAI$_{cor}$, the reclassification of diatom species identified a saproxenous species, *Cocconeis placentula var. placentula*, resulting in the larger YRDAI$_{cor}$ scores than the DAIpo, YRDAI$_{co-COD}$ and YRDAI$_{co-TP}$ scores. *Cocconeis placentula var. placentula*, which is one of the dominant species with a total number of 2672, accounted for about 22% of the identified diatoms in the watershed. The COD and TP optima values (3.95 mg/L, 0.03 mg/L) of *Cocconeis placentula var. placentula* were low, and *Cocconeis placentula* was known to be sensitive to pollutants [50]; it should be classified into the saproxenous group. However, it was still classified into the indifferent taxa by the coexistence-based approach, which is not reasonable. Here, the correlation-based approach performed better than the coexistence-based approach.

The ecological status of benthic diatoms is affected by many factors, and the factors may be location-specific [36]. Therefore, the application of the diatom index should be carefully verified to ensure that water quality is the main factor. Due to the considerable high variance for the biological indicators, the prolonged accumulation of investigative datasets is necessary. It is necessary to note that augmentability for the correlation-based approach is better than that for the coexistence-based approach, because the tolerance index scores of all diatoms need to be recalculated when adding new data. However, the correlation-based approach more easily discriminates diatom taxa.

It should be noted that a diatom community may change with time. Though some researchers found there was no significant difference in diatom index scores in different seasons [42,51], others held opposing opinions [42,52–54]. For example, the abundance of *Cocconeis placentula var. placentula* has a positive correlation with temperature [55]. Furthermore, due to the temporal and spatial change of a diatom community, the values of a regional diatom index may correspondingly change [56].

## 5. Concluding Remarks

Using the Yuqiao Reservoir watershed as an example, the establishment of a regional diatom index (YRDAI) based on the DAIpo index has been conducted via both the previous coexistence-based approach and the newly proposed correlation-based approach. The

localized diatom index (YRDAI) can better reflect the pollution level of the watershed of Yuqiao Reservoir than the borrowed index (DAIpo), indicating the applicability of both approaches. The correlation-based approach performed better than the coexistence-based approach in terms of calculation and augmentability. In particular, the correlation-based approach simplified the procedure of diatom reclassification in the establishment of a regional diatom index. In addition, the modified indices should be further improved through the continuous accumulation of datasets, and the proposed approaches can be applied to other watersheds as well. However, it is worthy to note that the regional diatom index cannot be directly compared due to its regional specificity.

**Author Contributions:** Conceptualization, X.L.; Methodology, J.F., X.L. and Y.L.; Software, Y.L.; Validation, Y.L.; Formal analysis, Y.L.; Investigation, J.F., Y.L. and S.Y.; Resources, P.M.; Data curation, Y.L. and S.Y.; Writing—original draft preparation, Y.L.; Writing—review and editing, X.L. and B.Z.; Visualization, Y.L.; Supervision, J.F. and X.L.; Project administration, J.F. and X.L.; Funding acquisition, J.F. and X.L. All authors have read and agreed to the published version of the manuscript.

**Funding:** This research was funded by National Social Science Fund (19AZD005), Tianjin Science and Technology Program (21YFSNSN00220), National Key Research & Development Program of China (2019YFE0122300) and Hebei Province Key Research & Development Program (22373301D).

**Data Availability Statement:** The data used to support the findings of this study are available from the corresponding author upon request.

**Acknowledgments:** We are grateful for the constructive comments from the anonymous reviewers.

**Conflicts of Interest:** The authors declare no conflict of interest.

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
