# Peer review of "How to Create a Regional Diatom-Based Index: Demonstration from the Yuqiao Reservoir Watershed, China"

_water, doi:10.3390/w14233926_

Round 1

Reviewer 1 Report (Previous Reviewer 2)

The Authors of the manuscript introduced comments as the Reviewers suggested.

I accept the revised version of the manuscript.

Author Response

We thank the reviewer for all the constructive comments.

Reviewer 2 Report (Previous Reviewer 5)

The Authors answered the previous reviewer's question.

Author Response

We thank the reviewer for all the constructive comments.

Reviewer 3 Report (New Reviewer)

The study was to produce YRDAI, an index of diatoms that appeared in the Yuqiao Reservoir basin in China, based on the correlation between COD and TP of pollutants in the field and the abundance of diatoms.

While known TDI and DAIpo used relatively abundant data, generalization of the index was somewhat insufficient by using data with a small scale or scale.

However, considering the characteristics of the diatom community with distinct regional characteristics, even the existing indices cannot overcome regional characteristics, so this attempt can be highly evaluated.

I would like to get answers to a few key questions.

1. The DAIpo index also uses BOD (the relationship between organic matter and bacteria) and classifies taxa into only three types, so it is virtually difficult to make a detailed classification according to the concentration gradient. It has a limitation in that it is difficult to make a relative comparison with the regional specificity.

Q - Why use COD rather than BOD? In what ways does it sound more rational and scientific?

2. Compared to the DAIpo index, the TDI index divides the pollution level more widely and gives weight to it to express a wide spectrum of pollution levels, and it is judged more reasonable than the TP index in that PO4 is used directly for growth of diatoms.

Q - Why did you use TP rather than PO4? In what ways does it sound more rational and scientific?

3. Various factors other than diatoms and some environmental factors are involved in the growth of diatoms. Many factors are already known. Please check the history of the Ditom index. A literature review is absolutely necessary. Many Chinese domestic scholars are also claiming this.

Q - How is the correlation with the existing Diatom indices?, What is the originality of the newly proposed index? Why is a regional diatom index required?

Round 2

Reviewer 3 Report (New Reviewer)

The authors do not seem to understand the intent of the questions in the reviews.

1. Although there is a high correlation between BOD and COD, it can be seen that it is necessarily appropriate to change the DAIpo index using BOD to COD. If the sensitivity or pollution level is adjusted using the relationship between diatoms and BOD appearing in a specific area, the pollution level can be sufficiently explained. The author must prove that COD is more effective than using BOD using experiments or more data than now.

2. TDI was evaluated by Kelly et al., similar to the authors, with sensitivity to PO4 and contamination level of diatoms that appeared in a specific area. Chinese ecologists (Wang et al.) are also developing more generalizable indices such as physiological characteristics, motility, and modified TDI using various parameters. Authors must prove that the DAIpo index can more effectively evaluate the pollution level using diatoms than the TDI index.

3. Recently, research on pollution level evaluation using diatom is very active. However, developing an index that is more effective than existing indexes is not an easy task. However, locality, individual differences, and selection of appropriate environmental parameters, which are difficult to explain in previous studies, require many challenges. Therefore, although the authors' attempts can be acknowledged in this paper, the novelity and differentiation from existing papers are not clear, and it is judged that it is difficult to express it as more efficient.

4. This mauscript must be revised starting with the title. Content such as "Development of a modified DAIpo index suitable for a specific region" should be included.

This manuscript is a resubmission of an earlier submission. The following is a list of the peer review reports and author responses from that submission.

Round 1

Reviewer 1 Report

How to create a regional diatom-based index: a case from the Yuqiao Reservoir watershed, China.

Reviewer's Report 

In this work, the authors tried to propose an index that could be used for assessing water quality. The proposed index is based on a Diatom Assemblage Index (DAIpo), published in 1986. The authors recommended their index as a regional diatom assemblage index (YRDAI) and used as a source of data a reservoir, probably in their area. The authors have carried out hard work regarding data collection in the field as well as laboratory analysis of the samples. However, I would like to express my reservations: (a) the whole procedure is rather complicated: water quality indices should be based on simple mathematical relations and limited number of variables. Complex procedures are rather cumbersome and increase the degree of uncertainty. (b) The "extent" of regionality in the present work is not clear to me and (c) from my personal experience, diatoms are rathe tolerant to environmental stress regarding pollution effects and therefore, an index based on diatoms could possibly be useful for rather serious forms of pollution. This index may be useful to cover grades (d) and (e) that is poor and bad status of the five grades of water quality status adopted by the European Union (the authors have mentioned in their text the EU Framework Directive 60/2000).

Furthermore, English-wise the text is not in an acceptable form for publication. I have indicated some corrections/ improvements in the introductory part of the manuscript, but if it is going to be published, the text should be thoroughly checked throughout.

Taking into account my comments, I am hesitant to suggest the submitted work for publication as it does not add, in my opinion, something new of particularly useful in the field of water quality indices.

Reviewer 2 Report

Reviewer’s comments

The manuscript titled “How to create a regional diatom-based index: A case form the Yuqiao Reservoir watershed, China” is important due to the fact that it demonstrate the regional diatom assemblage index based on a case study from the Yuqiao Reservoir watershed China. It also describes correlation between pollutant concentration and diatom abundance and compared with the traditional approach based on the coexistence index.

Specific comments:

Please, introduce more information about habitat, distribution, biology and ecology of study species.

How were all the diatom species taxonomic verifications performed?

Did the Authors use identification keys?

Please, read the paper by Abarca et al. Does the cosmopolitan diatom Gomphonema parvulum (Kutzing) Kutzing have a biogeography? PlosOne 2014, 9(1), e86885.

Do the Authors have photographic documentation of diatom?

Photographic documentation investigating in the Yuqiao Reservoir watershed is necessary.

I am positively surprised by the very detailed description of the methods used.

Figure 1 is scaled up.

Table 2 should also contain references from authors.

Reviewer 3 Report

The article is very interesting, we have plans to use these studies in our work.

Reviewer 4 Report

The paper presents an application of regional diatom assemblage index (YRDAI) for Yuqiao Reservoir. The subject addressed is within the scope of the journal. It is a topic of interest to the researchers in the related water quality assessment, but the paper only show a case, a very short time application, it is hard to say the results can reflect the whole story. My detailed comments are as follows:

1.         Unfortunately, the authors were sampled during the normal water season in mid-May 2019 only. We suggest authors add the statement about the repeatability of the diatom datasets. If the diatom index was useful in another seasons? How to proven that ?

2.         It is noted that your manuscript needs careful editing by someone with expertise in technical English editing paying particular attention to English grammar, spelling, and sentence structure so that the goals and results of the study are clear to the reader.

3.         In general, there is a lack of explanation of replicates and statistical methods used in the study. Furthermore, an explanation of why the authors did these various experiments should be provided in introduction section.

4.         The conclusions are overstated. For example, the study did not show if this diatom index is available in others season. I suggest rewritten the conclusions.

5.         A hypothesis needs to be presented。

6.         In current MS, we think its an application a diatom index in Yuqiao Reservoir, not “How to create a regional diatom-based index”.  Revised the title is necessary.

7.         Redraw the figure 3. Its unclear.

8.         In discussion section, delete the sentence (line 260- line 271). It is redundancy.

9.         In discussion section, “However, it was still classified into indifferent taxa by the coexistence-based approach, which is not reasonable. Here, the correlation-based approach performed better than the coexistence-based approach (line 302- line 304) ” , how to prove that ????

Reviewer 5 Report

Review on:How to create a regional diatom-based index: A case from the 2 Yuqiao Reservoir watershed, China

The aim of the article is to develop a regional diatom assemblage index (YRDAI) based on the data of Yuqiao Reservoir watershed. Although this is a specialized index for his region, the development steps could be interesting for broader application of diatom-based ecological assessment.

The major problem is that the data presented and the results are not in accordance. The values are presented in Table 1. and as an example, the correlation calculation is not valid.

Just for test I try cor(COD,RDAIco_COD) :-0.2310729 but in Table 3.  it is -0,65** but how ??? Was there a logarithmic transformation in the variable?

Also just in Figure 5 .

lm(formula = YRDAIco_COD ~ COD)

Multiple R-squared:  0.05339,    Adjusted R-squared:  0.02184
F-statistic: 1.692 on 1 and 30 DF,  p-value: 0.2032

Not R2=0,54 !!

It seems that not all data points remain in the final analyses ( in this case No 5. were excluded) This should be noted and explained.

Although removing No 5 point Multiple R-squared:  0.4172,    Adjusted R-squared:  0.3971
F-statistic: 20.76 on 1 and 29 DF,  p-value: 8.69e-05 not 0,54 as in the FIG3.

These issues should be addressed before further evaluation of the Manuscript.